Diversity of fish sound types in the Pearl River Estuary, China

Wang Zhi-Tao 1
Nowacek Douglas P. 2 3
Akamatsu Tomonari 4
Wang Ke-Xiong wangk@ihb.ac.cn 1
Liu Jian-Chang 5
Duan Guo-Qin 6
Cao Han-Jiang 6
Wang Ding wangd@ihb.ac.cn 1
1 The Key Laboratory of Aquatic Biodiversity and Conservation of the Chinese Academy of Sciences, Institute of Hydrobiology of the Chinese Academy of Sciences , Wuhan , P.R. China
2 Division of Marine Science and Conservation, Nicholas School of the Environment, Duke University of Marine Laboratory , NC , Beaufort , USA
3 Pratt School of Engineering, Duke University , Durham , NC , United States of America
4 National Research Institute of Fisheries Science, Fisheries Research and Development Agency , Kanagawa , Japan
5 Transport Planning and Research Institute, Ministry of Transport , Beijing , P.R. China
6 Hongkong-Zhuhai-Macao Bridge Authority , Guangzhou , China
Johnson Magnus
Electronic publication date: 2017 Oct 24
Publication date: 2017
Volume: 5
Electronic Location ID: e3924
Received 2017 May 23; Accepted 2017 Sep 23
Copyright: ©2017 Wang et al.
Copyright year: 2017
Copyright holder: Wang et al.
License: This is an open access article distributed under the terms of the Creative Commons Attribution License, which permits unrestricted use, distribution, reproduction and adaptation in any medium and for any purpose provided that it is properly attributed. For attribution, the original author(s), title, publication source (PeerJ) and either DOI or URL of the article must be cited.
License URL: https://creativecommons.org/licenses/by/4.0/

Keywords: Indo-pacific humpback dolphins, Hierarchical cluster analysis, Pearl River Estuary, Fish sound, Passive acoustic monitoring, Pulse train

Funding: National Natural Science Foundation of China 31070347 Ministry of Science and Technology of China 2011BAG07B05-3 Knowledge Innovation Program of the Chinese Academy of Sciences KSCX2-EW-Z-4 Special Fund for Agro-scientific Research in the Public Interest of the Ministry of Agriculture of China 201203086 State Oceanic Administration of China 201105011-3 NSFC 31170501 China Scholarship Council [2014]3026 Grants for this study were provided by the National Natural Science Foundation of China (NSFC, Grant No. 31070347), the Ministry of Science and Technology of China (Grant No. 2011BAG07B05-3), the Knowledge Innovation Program of the Chinese Academy of Sciences (Grant No. KSCX2-EW-Z-4) and the Special Fund for Agro-scientific Research in the Public Interest of the Ministry of Agriculture of China (Grant No. 201203086) to Ding Wang, the State Oceanic Administration of China (Grant No. 201105011-3) and NSFC (Grant No. 31170501) to Ke-Xiong Wang and the China Scholarship Council (Grant No. [2014]3026) to Zhi-Tao Wang. The funders had no role in study design, data collection and analysis, decision to publish, or preparation of the manuscript.

==============================
Background

Repetitive species-specific sound enables the identification of the presence and behavior of soniferous species by acoustic means. Passive acoustic monitoring has been widely applied to monitor the spatial and temporal occurrence and behavior of calling species.

Methods

Underwater biological sounds in the Pearl River Estuary, China, were collected using passive acoustic monitoring, with special attention paid to fish sounds. A total of 1,408 suspected fish calls comprising 18,942 pulses were qualitatively analyzed using a customized acoustic analysis routine.

Results

We identified a diversity of 66 types of fish sounds. In addition to single pulse, the sounds tended to have a pulse train structure. The pulses were characterized by an approximate 8 ms duration, with a peak frequency from 500 to 2,600 Hz and a majority of the energy below 4,000 Hz. The median inter-pulsepeak interval (IPPI) of most call types was 9 or 10 ms. Most call types with median IPPIs of 9 ms and 10 ms were observed at times that were exclusive from each other, suggesting that they might be produced by different species. According to the literature, the two section signal types of 1 + 1 and 1 + N10 might belong to big-snout croaker (Johnius macrorhynus), and 1 + N19 might be produced by Belanger’s croaker (J. belangerii).

Discussion

Categorization of the baseline ambient biological sound is an important first step in mapping the spatial and temporal patterns of soniferous fishes. The next step is the identification of the species producing each sound. The distribution pattern of soniferous fishes will be helpful for the protection and management of local fishery resources and in marine environmental impact assessment. Since the local vulnerable Indo-Pacific humpback dolphin (Sousa chinensis) mainly preys on soniferous fishes, the fine-scale distribution pattern of soniferous fishes can aid in the conservation of this species. Additionally, prey and predator relationships can be observed when a database of species-identified sounds is completed.

Introduction

The Pearl River Estuary (21°40′–22°50′N; 112°50′–114°30′E) is in a subtropical area of the northern South China Sea. The estuary is one of the most economically developed regions in China, and the rapid local industrialization and large-scale infrastructure projects, e.g., the ongoing construction of the Hong Kong-Zhuhai-Macao bridge (Wang et al., 2014b) and the Guishan wind farm project (Wang et al., 2015b), have placed an extraordinarily heavy burden on coastal environments and accelerated human damage to coastal ecosystems.

Sound production in soniferous fish has been shown to be associated with reproduction (e.g., courtship and spawning) and territorial or aggressive behavior (Hawkins & Amorim, 2000; Takemura, Takita & Mizue, 1978). Most of the repetitive fish sounds are species specific (Tavolga, 1964), which enables the identification of the distribution and behavior of soniferous species by acoustic means. As a noninvasive technology, passive acoustic monitoring has been widely applied to map the spatial (over a wide range of habitats and at varied depths) (Wall, Lembke & Mann, 2012; Wall et al., 2013) and temporal (diel, seasonal and annual) (Locascio & Mann, 2011; Ruppé et al., 2015; Turnure, Grothues & Able, 2015) occurrence and behavior of soniferous fishes, even in severe conditions.

Overfishing and ocean pollution in the past decade have led to a dramatic decrease in fish in the wild fisheries of China (Liu & Sadovy, 2008; Sadovy & Cheung, 2003). The endemic species of giant yellow croaker (Bahaba taipingensis), which is highly valued as a traditional medicine of its swim bladder and was an important fish stock before the 1960s, collapsed in the wild and was determined to be commercially extinct in 1997 (Sadovy & Cheung, 2003). The spotted drum (Protonibea diacanthus) and large yellow croaker (Larimichthys crocea, which is endemic to East Asia and was once one of the three top commercial marine fishes in China), have been severely depleted throughout their geographic range since the 1980s and have now almost entirely disappeared from landings (Liu & Sadovy, 2008; Sadovy & Cheung, 2003). The most recent study of Indo-Pacific humpback dolphins (Sousa chinensis, locally called the Chinese white dolphin) biosonar activity in the Pearl River Estuary indicated that its diel, seasonal and tidal patterns might be ascribed to the spatial–temporal variability of its prey (Wang et al., 2015b); however, little attention has been paid to local fishes, with only sporadic fishery distribution data with poor temporal and spatial resolution obtained from 1986 to 1987 by bottom trawl and in 1998 by beam trawl and hang trawl (Li, Chen & Sun, 2000; Wang & Lin, 2006). The fine-scale distribution pattern of humpback dolphin prey has yet to be investigated.

In this study, the ambient biological sounds in the Pearl River Estuary were recorded using passive acoustic monitoring. Suspected fish sounds were quantitatively and qualitatively characterized. We compared the species-specific sounds thorough a literature review, especially of those species that are distributed in the research area, to confirm the caller’s identity. These baseline data can serve as a first step toward mapping the spatial and temporal distribution patterns of soniferous fishes in the estuary. Moreover, they are helpful for planning fisheries management and evaluation of the damage to aquatic environments (e.g., spawning grounds of the sciaenids) from various large-scale infrastructure projects because marine environmental impact assessments must be based upon a good understanding of the local baseline biodiversity. Additionally, the baseline data can aid in the protection of local humpback dolphins and the implementation of conservation strategies.

Methods

Acoustic data recording system

Underwater acoustic recordings were made using a Song Meter Marine Recorder (Wildlife Acoustics, Inc., Maynard, MA, USA), which included an HTI piezoelectric omnidirectional hydrophone (model HTI-96-MIN; High Tech, Inc., Long Beach, MS, USA) with a sensitivity of −164 dB re 1 V/µPa at 1 m distance, a recording bandwidth of 2 Hz–48 kHz and a flat frequency response over a wide range of 2 Hz–37 kHz (±3 dB). The hydrophone also included a programmable autonomous signal processing unit integrated with a band-pass filter and a pre-amplifier. The signal processing unit can log data at a resolution of 16 bits and at a 96 kHz sampling rate, with a storage capacity of 512 GB. The signal processing unit was sealed inside a waterproof PVC housing and was submersible to 150 m. The recording system was calibrated prior to shipment from the manufacturer.

Data collection

Static acoustic monitoring was conducted underwater at the base of a telephone signal tower (22°07′54″N, 113°43′54″E) located among the Sanjiao, Chitan and Datou islands (Fig. 1). The recordings were taken continuously throughout deployment periods from May 26 to June 4, 2014, and June 17 to 22, 2014, at a 96 kHz sampling rate. The acoustic recording system was attached to a steel wire rope and suspended below the signal tower in the middle of water column 4.0 m above the ocean floor and approximately 3.0–5.8 m (depending on the tide conditions) below the water surface. A 40 kg anchor block was attached on the bottom of the steel wire rope and laid down on the seabed to reduce the movement of the recording system due to water currents.

Figure 1 Map of the passive acoustic monitoring area.

Acoustic data analysis

Upon retrieval of the recorder, the acoustic data were downloaded and processed. Raven Pro Bioacoustics Software (version 1.4; Cornell Laboratory of Ornithology, NY, USA) was used to initially visualize the acoustic data in the spectrogram (window type: Hann windows; fast Fourier transform (FFT) size: 2048 samples; frame overlapping: 80%; frequency grid spacing: 46.88 Hz; temporal grid resolution: 4.26 ms). Only calls with good signal-to-noise ratios (SNR >15 dB, noise level obtained just before or after the pulse) and satisfying the criteria of no interference by other sounds were extracted for further quantitative analyses. To make the data more independent and reduce the possibility of using multiple sounds from the same individual, only one signal was extracted for each call type in every 10 min bin for further analysis.

Figure 2 Schematic diagram of the signal analysis.

(A) Oscillogram of the raw data with seven pulses. (B) Pulses detected by the pulse-peak detector. Vertical dashed lines denote the starting (green), peak (red), and ending (blue) points of a pulse. (C) Close-up of the oscillogram of extracted 8 ms pulses showing the fine-scale call structure. (D) The cumulative energy of the extracted pulse, τ95%, was the duration containing 95% of the cumulative energy of the pulse, which was derived from the time difference between the 2.5th and 97.5th cumulative energy percentiles. (E) Normalized signal envelope of the extracted pulse; τ−3 dB and τ−10 dB are the time differences between the −3 dB and −10 dB end points relative to the peak amplitude of the signal envelope, respectively. (F) Normalized power spectrum of the extracted pulse. Spectrum configuration: FFT size, 96,000; frequency grid spacing, 1 Hz.

The recorded sounds generally featured single or multiple-pulse structures. A custom acoustic analysis routine based on MATLAB 7.11.0 (The Mathworks, Natick, MA, USA) was developed to analyze the extracted calls. For each call, the peak amplitude time for each pulse within the call was logged using a pulse-peak detector. Through trial and error, the pulse was defined and extracted as an 8 ms signal that began 2.5 ms before and ended 5.5 ms after the time point of the peak amplitude (Figs. 2B and 2C). The 8 ms definition was validated because it encompassed the majority of the energy of a pulse and was longer than the shortest interval between pulses within a call. The sonic parameters of the number of pulses in a call, total call duration (in ms), inter-pulsepeak interval (IPPI), and the inter-pulse interval (IPI) were calculated for each call. Call duration is derived by adding 8 ms to the time difference of the last pulsepeak and the first pulsepeak; IPPI is the time difference between the peak amplitude of consecutive pulse units in the train, which is equal to the pulse period in the literature (Parmentier et al., 2009), and IPI is the time interval between the end of one pulse and the onset of the next one in a series. The temporal characteristics for each 8 ms pulse were computed as τ95%, τ−3dB and τ−10dB.τ95% is the duration containing 95% of the cumulative energy of the pulse (Fig. 2D), which began when 2.5% of the cumulative signal energy was reached (CE2.5% in Fig. 2D) and ended when 97.5% of the cumulative signal energy was reached (CE97.5% in Fig. 2D), and τ−3dB and τ−10dB are the time differences between the end points that were 3 dB and 10 dB lower than the peak amplitude of the envelope of the pulse waveform, respectively (Fig. 2E). The signal envelope was generated by taking the absolute value of the waveform after applying the Hilbert transform function (Au, 1993; Madsen & Wahlberg, 2007). The frequency and bandwidth properties for each 8 ms pulse were determined from the power spectrum, which was calculated from the squared fast Fourier transform of a 96,000-point Hanning window. Parameters of the peak frequency (fp, the frequency at which the spectrum has its maximum value) (Fig. 2F), center frequency (fc, the frequency that divides the power spectrum into equal energy halves) and centralized root-mean-square bandwidth (BWrms, the spectral standard deviation of the fc of the spectrum) (Au, 1993; Madsen & Wahlberg, 2007) were measured since they were proposed to be good descriptive parameters for signals with bimodal spectra (Au, 2004). Parameters of 3-dB and 10-dB bandwidths were not measured since they might only cover the frequency range near the peak frequency and tend to provide a misrepresentation of the bandwidth of signals with bimodal spectra (Au, 2004). The quality factor of each pulse (Q, an appropriate way to define the relative width of a signal) was computed as the ratio of the fc to the BWrms (Au, 1993; Au, 2004). The sound pressure levels (SPLs, dB re 1 µPa) and energy flux density (EFD, dB re 1 µPa2s) were derived for each 8 ms pulse over its τ95%. The SPL parameters included the zero-to-peak SPL (SPLzp) and the root-mean-square SPL (SPLrms) (Urick, 1983). The absolute pressure levels were derived by subtracting the sensitivity of the hydrophone and the gain due to the amplifier (Urick, 1983).

The pooled distribution pattern of the IPPI for all analyzed calls was characterized by a multi-peak mode, with a distribution curve peaking at 9, 10, 12, 13 and 18 ms (Fig. 3A). Previous experience in fish acoustic analysis by other investigators indicated that the IPPI was the most reliable basis for signal identification and species-specific recognition (Mann & Lobel, 1997; Parmentier et al., 2009; Spanier, 1979), and most signals in our database ended with a pulse train featuring regular IPPIs (Table 1). In this study, calls were classified into types primarily based on their IPPI patterns and their amplitude and temporal modulation patterns (Table 1). The calls were initially grouped according to the number of sections they contained (Table 1). For each call, pulses with IPPIs greater than 1.5 times the median IPPI of the call were divided into different sections. Based on the bimodal distribution of the IPPI for calls that consisted of fewer than three pulses, pulses with an IPPI greater than 24 ms (three times the duration of a single pulse of 8 ms) were divided into different sections (Fig. 3B). To name each call type, such as 2 + 1 + N10, (1 − )4 + (2 − )2 + N10 and iN13 (Figs. 4–6, Figs. S1–S26), ‘+’ was used to separate the different sections of a call, a number was used to denote the number of pulse for that section and ‘(1 −)’ and ‘(2−)’ to denote repeated sections that consist of one or two pulses, respectively, with digital superscripts denoting the number of repeats in a repeating section. ‘N’ was used to denote the last section of a call with a variable number of pulses, and the digital subscripts denote the median IPPIs of the last portion of the call; the subscript i was used to denote calls with a zero-to-peak sound pressure level of the first pulse approximately 10 dB weaker than that of the remainder of the call. Occasionally, a train of calls was extracted with significantly higher SNR (SNR > 25 dB), a regular inter-call interval, and a gradually changing pattern in its sound pressure level distinct from the ambient biological sounds. These sounds were likely produced by the same individual fish, which facilitated the estimation of the inter-call intervals.

Figure 3 Distribution pattern of the inter-pulsepeak interval (IPPI) for all analyzed calls (A) and call types with fewer than three pulses (B).

The distribution pattern of the pooled IPPIs peaked at 9, 10, 12,13 and 18 ms (inset figure in A). Call types with fewer than three pulses, including a two-pulse call in the 2, 1 + 1, 1 + N19, and iN13 call types and a three-pulse call in the iN13, N13, N17, and (1 − )2 + N10 call types. The bimodal distribution of the IPPI (inset figure in B) validated the selection of 24 ms, three times the duration of a single 8 ms pulse, as a threshold for dividing pulses of a call into different sections. The insets show magnified time scales of the IPPI for 8–20 ms and 10–52 ms.

Table 1 Call type classification.

Type	Call name	No. of sections	Inter-pulsepeak interval (IPPI) pattern	Observed No. of pulses in section N	
1	1	One			
2	2	One	IPPIs converged at 13 ms		
3	N9	One	Decreasing then increasing IPPI, median at 9 ms	29–30, 33–37	
4	N10	One	Decreasing then increasing IPPI, median at 10 ms	27–29, 33–36, 43, 45, 51	
5	N13	One	Nearly constant IPPI at 13 ms	3–7, 9, 11, 12, 14	
6	N17	One	Increasing IPPI, median at 17 ms	3–15,18	
7	iN13	One	Increasing, decreasing, then increasing IPPI, median at 13 ms	2–5, 9–17	
8	iN15	One	Decreasing IPPI, median at 15 ms	7–11, 13, 15	
9	1 + 1	Two	IPPI median at 41 ms		
10	1 + N10	Two	Nearly constant IPPI, median at 10 ms	7–13, 15–25, 27, 28	
11	1 + N12	Two	Nearly constant IPPI, median at 12 ms	13–26	
12	1 + N19	Two	Increasing IPPI, median at 19 ms	2–8, 10, 11	
13	2 + N9	Two	Near constant IPPI, median at 9 ms	23, 25, 27, 28, 30	
14	2 + N10	Two	Near constant IPPI, median at 10 ms	19, 26, 27	
15	2 + N18	Two	Increasing IPPI, median at 18 ms	3–8, 10	
16	3 + N9	Two	Near constant IPPI, median at 9 ms	24–26, 29, 30	
17	3 + N10	Two	Near constant IPPI, median at 10 ms	3–11, 24–25, 27–34, 37–39, 44	
18	3 + N17	Two	Increasing IPPI, median at 17 ms	4–7	
19	4 + N9	Two	Near constant IPPI, median at 9 ms	25–27, 31	
20	4 + N10	Two	Near constant IPPI, median at 10 ms	3–7, 15, 25, 28, 30–31, 33, 35, 36	
21	4 + N17	Two	Increasing IPPI, median at 17 ms	6	
22	5 + N10	Two	Nearly constant IPPI, median at 10 ms	3–5, 7	
23	(1 − )2 + N9	Three	Nearly constant IPPI, median at 9 ms	19, 22, 23	
24	(1 − )2 + N10	Three	Nearly constant IPPI, median at 10 ms	2, 9–24, 29, 30	
25	(1 − )2 + N12	Three	Nearly constant IPPI, median at 12 ms	6–11, 13–15, 19–21	
26	1 + 2 + N10	Three	Nearly constant IPPI, median at 10 ms	16	
27	1 + 2 + N18	Three	Nearly constant IPPI, median at 18 ms	5, 7	
28	2 + 1 + N9	Three	Nearly constant IPPI, median at 9 ms	21, 23–25, 28, 29, 31, 32	
29	2 + 1 + N10	Three	Nearly constant IPPI, median at 10 ms	23, 25–28, 30, 32, 34, 35, 40	
30	(2 − )2 + N10	Three	Nearly constant IPPI, median at 10 ms	23, 26	
31	3 + 1 + N9	Three	Nearly constant IPPI, median at 9 ms	23–25, 27, 30–32, 34	
32	3 + 1 + N10	Three	Nearly constant IPPI, median at 10 ms	27–31, 33–35, 37	
33	3 + 2 + N9	Three	Nearly constant IPPI, median at 9 ms	26	
34	4 + 1 + N10	Three	Nearly constant IPPI, median at 10 ms	21, 29–31, 33	
35	(1 − )3 + N9	Four	Nearly constant IPPI, median at 9 ms	18, 21, 26, 29	
36	(1 − )3 + N10	Four	Nearly constant IPPI, median at 10 ms	1, 9–14, 16, 17, 19, 23–25, 27–29, 31, 33	
37	(1 − )3 + N12	Four	Nearly constant IPPIs, median at 12 ms	8, 10, 13	
38	(1 − )2 + 2 + N9	Four	Nearly constant IPPI, median at 9 ms	26, 29	
39	(1 − )2 + 2 + N10	Four	Nearly constant IPPI, median at 10 ms	20, 21, 29	
40	(1 − )2 + 3 + N10	Four	Nearly constant IPPI, median at 10 ms	18	
41	2 + (1 − )2 + N9	Four	Nearly constant IPPI, median at 9 ms	22, 23	
42	2 + (1 − )2 + N10	Four	Nearly constant IPPI, median at 10 ms	20–24, 26–33, 36	
43	2 + 1 + 2 + N9	Four	Nearly constant IPPI, median at 9 ms	28	
44	2 + 1 + 2 + N10	Four	Nearly constant IPPI, median at 10 ms	22, 25, 30	
45	3 + (1 − )2 + N9	Four	Nearly constant IPPI, median at 9 ms	25	
46	(1 − )4 + N9	Five	Nearly constant IPPI, median at 9 ms	15, 18, 23, 24	
47	(1 − )4 + N10	Five	Nearly constant IPPI, median at 10 ms	1, 6, 7, 11, 13, 16–25, 27, 28	
48	(1 − )4 + N12	Five	Nearly constant IPPI, median at 12 ms	11	
49	(1 − )3 + 2 + N10	Five	Nearly constant IPPI, median at 10 ms	20, 21	
50	(1 − )3 + 3 + N10	Five	Nearly constant IPPI, median at 10 ms	17	
51	(1 − )2 + 2 + 1 + N10	Five	Nearly constant IPPI, median at 10 ms	26	
52	(1 − )2 + 2 + 3 + N10	Five	Nearly constant IPPI, median at 10 ms	14	
53	2 + (1 − )3 + N10	Five	Nearly constant IPPI, median at 10 ms	23–25, 27, 28, 32	
54	(1 − )5 + N9	Six	Nearly constant IPPI, median at 9 ms	17, 21	
55	(1 − )5 + N10	Six	Nearly constant IPPI, median at 10 ms	1, 16–23, 26	
56	(1 − )4 + 2 + N10	Six	Nearly constant IPPI, median at 10 ms	15, 18–20, 28	
57	(1 − )4 + 3 + N11	Six	Nearly constant IPPI, median at 11 ms	11	
58	(1 − )3 + 2 + 1 + N10	Six	Nearly constant IPPI, median at 10 ms	16, 18	
59	2 + (1 − )4 + N10	Six	Nearly constant IPPI, median at 10 ms	22	
60	(1 − )6 + N10	Seven	Nearly constant IPPI, median at 10 ms	14–17, 19, 20, 24	
61	(1 − )5 + 2 + N10	Seven	Nearly constant IPPI, median at 10 ms	16–18	
62	(1 − )5 + 3 + N10	Seven	Nearly constant IPPI, median at 10 ms	16	
63	(1 − )4 + 2 + 1 + N10	Seven	Nearly constant IPPI, median at 10 ms	16	
64	(1 − )4 + (2 − )2 + N10	Seven	Nearly constant IPPI, median at 10 ms	20	
65	(1 − )7 + N10	Eight	Nearly constant IPPI, median at 10 ms	11, 13, 14, 19, 21	
66	(1 − )5 + (2 − )2 + N10	Eight	Nearly constant IPPI, median at 10 ms	9, 15	
Notes.

For each signal, pulses with an inter-pulsepeak interval (IPPI) greater than 1.5 times the median IPPI of the signal were grouped into different sections. For signals that consisted of fewer than three pulses, pulses with an IPPI greater than 24 ms (three times the duration of a single pulse) were further grouped into different sections. In the call name column, ‘+’ is used to separate different sections of a call; the number denotes the number of pulses in that section; ‘(1 − )’ and ‘(2 − )’ denote repeated sections that consist of one and two pulses, respectively; the digital superscripts denote the number of repeats in the repeating section; ‘N’ denotes the last section of a call that varied in the number of pulses; the digital subscripts denote the median IPPIs of the last portion of the call; the subscript i denotes calls with a zero-to-peak sound pressure level of the first pulse approximately 10 dB weaker than that of the remainder within the call. For call types with more than one portion, the IPPI pattern of the last section is given.

Figure 4 Characteristic of the N9 (first column), N10 (second column), N13 (third column), and N17 (fourth column) call types.

Row 1 (A–D) and row 2 (E–H) are the oscillogram and sonogram, respectively, of a representative signal for each call type. Row 3 (I–L) is the duration of a call as a function of the number of pulses within the call for each call type. Results of the pooled inter-pulsepeak interval (M–P in row 4), sound pressure level (Q–T in row 5), peak frequency (U–X in row 6), and center frequency (Y–BB in row 7) of each pulse versus the order at which it occurs within a call for each call type are also given. For the boxplot, the line inside the box indicates the median value, and the upper and lower box borders are the first and third quartiles, respectively. The length of the box is the interquartile range (IQR). The whiskers extend to the most extreme data within the limit of 1.5 IQRs from the end of the box. Open circles (o) denote mild outliers with values greater than 1.5 IQRs but fewer than 3 IQRs from the end of the box. Asterisks (*) denote extreme outliers with values greater than three box lengths from the upper or lower edges of the box. Sonogram configuration: FFT size, 96,000; window type, Hanning; overlap samples per frame, 95%.

Figure 5 Characteristics of the iN13 (first column) and iN15 (second column) call types.

Row 1 (A–B) and row 2 (C–D) are the oscillogram and sonogram, respectively, of a representative signal for each call type. Row 3 (E–F) is the duration of a call as a function of the number of pulses within the call for each call type. Results of the pooled inter-pulsepeak interval (G and H in row 4), sound pressure level (I and J in row 5), peak frequency (K and L in row 6), and center frequency (M and N in row 7) of each pulse versus the order at which it occurs within a call for each call type are also given.

Figure 6 Characteristics of the 1 + N10 (first column), 1 + N12 (second column) and 1 + N19 (third column) call types.

Row 1 (A–C) and row 2 (D–F) are the oscillogram and sonogram, respectively, of a representative signal for each call type. Row 3 (G–I) is the duration of a call as a function of the number of pulses within the call. Results of the pooled inter-pulsepeak interval (J–L in row 4), sound pressure level (M–O in row 5), peak frequency (P–R in row 6), and center frequency (S–U in row 7) of each pulse versus the order at which it occurs within a call for each call type are also given.

Statistical analysis

Descriptive statistics were used to summarize the biographical information. All the parameters were tested for normality (using the Shapiro–Wilk test for data sets <50 or the Kolmogorov–Smirnov test for data sets ≥50) and homoscedasticity (using Levene’s test for equality of variance) (Zar, 1999). Because of the grossly skewed distribution of the majority of the data, the descriptive parameters of median, quartile deviation (QD), 5th percentile (P5), and 95th percentile (P95) were adopted. The QD was defined as one-half the interquartile range, which is the difference between the 25th and 75th percentiles in a frequency distribution.

Principal component analysis was used to identify the variables explaining the most variance among the acoustic parameters. Call types with an analyzed number greater than five were extracted for further discriminant and cluster analyses. Canonical discriminant analysis was used to assess the variation among call types relative to the variation within call types and determine the validity of our call types. Hierarchical cluster analysis (Romesburg, 2004), a step-wise process that merges the two closest or furthest data points at each step and builds a hierarchy of clusters based on the distance between them, was applied to discover similar call types in each set. Because the amplitude parameters were not critical for species recognition (Ha, 1973) and the call duration was dependent on the number of pulses in a call (Parmentier et al., 2009), these parameters were not included in the principal component analysis, canonical discriminant analysis and hierarchical cluster analysis. The statistical analyses were performed using Statistical Package for the Social Sciences 16.0 for Windows (SPSS Inc., Chicago, IL, USA).

Results

Ambient biological sounds and suspected fish sounds were recorded over all the 16 recording days and sometimes formed dense choruses of individual sound emissions produced simultaneously and/or overlapping with each other that obscured the signals and could not be discriminated individually, especially before dusk. In addition to some single pulses, individual calls tended to possess a multi-pulse burst structure. The most representative pulse consisted of six oscillations (Fig. 2C). Owing to the single hydrophone methodology, animal localization was not possible in this study. The recorded sound was occasionally clipped, indicating that the source level of the sound was higher than 164 dB (limited by the hydrophone sensitivity). A total of 1,408 calls comprising 18,942 pulses were extracted for statistical analysis and were categorized into 66 call types (Table 1).

Single-section calls

Calls that consisted of a single section included call types 1, 2 (Table S1, Fig. S1), N9, N10, N13, N17 (Table 2, Fig. 4), iN13 and iN15 (Table 3, Fig. 5).

Table 2 Descriptive statistics of sonic parameters of the N9, N10, N13, and N17 call types.

		Dur	IPPI	τ95%	τ−3 dB	τ−10 dB	fp	fc	BWrms	Q	SPLzp	SPLrms	EFD	N1	N2	N3	
N9	P50	300.30	9.09	3.22	0.31	0.36	856	1,366	1,228	1.14	130.99	122.81	147.51	9	287	296	
	QD	28.03	0.25	0.48	0.10	0.21	59	153	557	0.32	2.50	3.34	2.97				
	P5	253.39	8.32	2.42	0.15	0.16	747	1,015	679	0.48	122.99	112.08	139.48				
	P95	334.04	9.49	6.49	1.24	1.53	1,144	2,273	4,709	1.62	136.98	128.21	152.82				
N10	P50	356.94	10.50	4.35	0.21	1.16	903	1,580	1,222	1.27	139.67	128.22	154.66	13	448	461	
	QD	59.78	0.29	1.51	0.11	0.48	113	289	525	0.31	9.20	10.27	9.09				
	P5	275.72	9.73	2.93	0.11	0.15	667	1,024	772	0.62	123.93	110.66	138.54				
	P95	544.98	11.07	7.39	0.43	1.72	1,274	2,450	3,705	1.80	147.13	137.36	162.00				
N13	P50	119.15	13.11	3.33	0.39	0.86	1,296	1,776	702	2.53	156.35	146.42	170.87	26	190	216	
	QD	46.27	0.22	0.48	0.02	0.09	139	44	66	0.23	1.33	1.45	1.16				
	P5	35.06	12.67	2.54	0.34	0.72	1,178	1,681	595	1.23	150.66	140.18	166.38				
	P95	170.20	13.93	5.99	0.48	1.19	2,390	1,931	1,548	2.92	158.05	147.96	172.61				
N17	P50	149.11	17.44	4.40	0.52	0.97	789	1,144	490	2.35	159.56	151.11	177.30	462	3,803	4,265	
	QD	10.00	1.11	0.34	0.02	0.05	49	48	27	0.11	1.48	1.36	1.41				
	P5	141.53	16.04	4.02	0.50	0.93	765	1,100	464	2.23	158.17	149.75	175.99				
	P95	179.74	19.31	5.42	0.64	1.82	957	1,278	641	2.65	163.93	155.10	181.30				
Notes.

P50 median; P5 and P95, 5th percentile and 95th percentile, respectively

QD quartile deviation

Dur duration

IPPI inter-pulsepeak interval

τ95% duration of 95% cumulative energy

τ−3 dB and τ−10 dB duration of −3 dB and −10 dB of the peak amplitude of the enveloped signal, respectively

fp peak frequency

fc center frequency

BWrms centralized root-mean-square bandwidth

Q quality factor

SPLzp and SPLrms zero-to-peak and root-mean-square sound pressure levels, respectively

EFD energy flux density

N1, N2 and N3 number of calls, inter-pulsepeak intervals and pulses analyzed, respectively

The duration is in seconds, the frequency is in Hz, the SPL is in dB re 1 µPa, and the EFD is in dB re 1 µPa2s. The IPIs are not shown here and can be obtained by subtracting 8 ms from the IPPIs. The same notation was used for the following tables.

Table 3 Descriptive statistics of sonic parameters of the iN13 and iN15 call types.

		Dur	IPPI	τ95%	τ−3 dB	τ−10 dB	fp	fc	BWrms	Q	SPLzp	SPLrms	EFD	N1	N2	N3	
iN13	P50	174.10	13.15	3.17	0.39	0.82	1,490	1,770	663	2.66	157.38	147.01	171.91	111	1,266	1377	
	QD	17.49	0.35	0.42	0.03	0.13	217	49	52	0.22	2.09	2.05	1.91				
	P5	33.26	12.35	2.42	0.33	0.45	1,184	1,601	545	1.54	146.21	135.78	162.38				
	P95	202.23	15.37	5.75	0.60	1.31	2,390	1,930	1,038	3.29	161.03	151.31	175.66				
iN15	P50	169.31	14.96	3.12	0.41	0.42	1,510	1,787	929	1.95	142.26	133.21	157.60	16	158	174	
	QD	19.04	1.51	0.33	0.10	0.15	167	47	122	0.22	2.89	2.47	2.69				
	P5	139.67	13.55	2.70	0.24	0.20	1,283	1,750	823	1.70	140.50	131.32	155.86				
	P95	192.87	19.30	5.30	0.57	0.65	2,202	2,362	2,059	2.98	152.37	143.35	167.28				

Two-section calls

Calls consisting of two sections included call types 1 + 1 (Table S1, Fig. S1), 1 + N10, 1 + N12, 1 + N19 (Table 4, Fig. 6), 2 + N9, 2 + N10, 2 + N18 (Table S2, Fig. 7 and Fig. S2), 3 + N9, 3 + N10, 3 + N17 (Table S3, Fig. 7 and Fig. S3), 4 + N9, 4 + N10, 4 + N17 (Table S4, Fig. 7 and Fig. S4), and 5 + N10 (Table S5, Fig. S5).

Table 4 Descriptive statistics of sonic parameters of the 1 + N10, 1 + N12 and 1 + N19 call types.

		Dur	IPPI	τ95%	τ−3 dB	τ−10 dB	fp	fc	BWrms	Q	SPLzp	SPLrms	EFD	N1	N2	N3	
1 + N10	P50	232.80	10.15	3.42	0.41	1.08	1,128	1,474	669	2.12	152.67	143.04	167.93	75	1,432	1,507	
	QD	22.34	0.18	0.59	0.04	0.42	144	122	84	0.30	3.43	3.29	3.50				
	P5	124.18	9.82	2.20	0.33	0.38	792	1,148	550	0.97	141.26	132.09	157.57				
	P95	278.07	27.17	6.19	0.58	1.56	1,355	1,708	1,385	2.80	161.00	150.70	175.61				
1 + N12	P50	260.67	11.73	3.30	0.40	0.43	879	1,213	684	1.67	138.77	130.44	155.31	15	292	307	
	QD	41.74	0.19	0.64	0.05	0.25	41	130	227	0.48	7.49	6.98	6.34				
	P5	183.67	11.55	2.23	0.19	0.20	796	935	525	0.67	122.02	112.12	138.95				
	P95	337.81	35.09	5.44	0.90	1.35	1,193	1,516	2,284	2.34	154.90	144.12	170.29				
1 + N19	P50	165.96	18.73	4.64	0.52	1.01	789	1,105	480	2.33	157.80	149.44	175.92	105	591	696	
	QD	14.61	0.99	0.36	0.03	0.13	42	62	33	0.16	2.05	2.20	2.12				
	P5	115.74	15.75	3.71	0.49	0.89	722	898	395	1.15	144.06	135.10	163.23				
	P95	195.68	79.77	6.87	0.79	3.04	946	1,254	895	2.61	162.68	153.89	180.29				

Figure 7 Representative oscillogram and sonogram of two section signals with the first section contain two pulses (2 + N9 in A and D and 2 + N18 in G and J), three pulses (3 + N9 in B and E and 3 + N17 in H and K) and four pulses (4 + N9 in C and F and 4 + N17 in I and L).

Oscillograms in row 1 (A–C) and the corresponding sonograms in row 2 (D–F) are call types with IPPIs medians at 9 ms, whereas oscillograms in row 3 (G–I) and its corresponding sonograms in row 4 (J–L) are call types with IPPIs medians at 17 ms.

Three-section calls

Calls consisting of three sections included call types (1 − )2 + N9, (1 − )2 + N10, (1 − )2 + N12 (Table S6, Fig. 8 and Fig. S6), 1 + 2 + N10, 1 + 2 + N18 (Table S7, Fig. S7), 2 + 1 + N9, 2 + 1 + N10 (Table S8, Fig. S8), (2 − )2 + N10 (Table S9, Fig. S9), 3 + 1 + N9, 3 + 1 + N10 (Table S10, Fig. S10), 3 + 2 + N9 (Table S11, Fig. S11) and 4 + 1 + N10 (Table S9, Fig. S9).

Figure 8 Reprsentative oscillogram and sonogram of the (A and D) (1 − )2 + N10, (B and E) (1 − )3 + N10, (C and F) (1 − )4 + N10 (G and J) (1 − )5 + N10, (H and K) (1 − )6 + N10, and (I and L) (1 − )7 + N10 call types.

Four-section calls

Calls consisting of four sections included call types (1 − )3 + N9, (1 − )3 + N10, (1 − )3 + N12 (Table S12, Fig. 8 and Fig. S12), (1 − )2 + 2 + N9, (1 − )2 + 2 + N10 (Table S13, Fig. S13), (1 − )2 + 3 + N10 (Table S14, Fig. S14), 2 + (1 − )2 + N9, 2 + (1 − )2 + N10 (Table S15, Fig. S15), 2 + 1 + 2 + N9, 2 + 1 + 2 + N10 (Table S16, Fig. S16) and 3 + (1 − )2 + N9 (Table S11, Fig. S11).

Five-section calls

Calls consisting of five sections included call types (1 − )4 + N9, (1 − )4 + N10, (1 − )4 + N12 (Table S17, Fig. 8C and Fig. S17), (1 − )3 + 2 + N10, (1 − )3 + 3 + N10 (Table S18, Fig. S18), (1 − )2 + 2 + 1 + N10, (1 − )2 + 2 + 3 + N10 (Table S19, Fig. S19), and 2 + (1 − )3 + N10 (Table S20, Fig. S20).

Six-section calls

Calls consisting of six sections included call types (1 − )5 + N9, (1 − )5 + N10 (Table S21, Fig. 8 and Fig. S21), (1 − )4 + 2 + N10, (1 − )4 + 3 + N11 (Table S22 and Fig. S22), (1 − )3 + 2 + 1 + N10 (Table S23 and Fig. S23), and 2 + (1 − )4 + N10 (Table S20, Fig. S20).

Seven-section calls

Calls consisting of seven sections included call types (1 − )6 + N10 (Table S24, Fig. 8H and Fig. S24), (1 − )5 + 2 + N10, (1 − )5 + 3 + N10 (Table S25 and Fig. S25), (1 − )4 + 2 + 1 + N10 (Table S23 and Fig. S23), and (1 − )4 + (2 − )2 + N10 (Table S26 and Fig. S24).

Eight-section calls

Calls consisting of eight sections included call types (1 − )7 + N10 (Table S24, Fig. 8I and Fig. S24) and (1 − )5 + (2 − )2 + N10 (Table S26 and Fig. S26).

Principal component, discriminant function and hierarchical cluster analyses

The principal component analysis indicated that approximately 81.1% of the variability is explained by the first four principal components (39.2% by principal component 1, 18.1% by principal component 2, 13.2% by principal component 3, and 10.6% by principal component 4). Principal component 1 was loaded with the τ−3 dB, τ−10 dB, fc, BWrms and Q parameters. Principal component 2 was loaded with fp. The third component describes the temporal parameter of the IPPI, and the fourth component describes the temporal parameters of τ−10 dB and the IPPI. The validity of our call types was confirmed using a canonical discriminant function that grouped N17, 1 + N19, 2 + N18 and 3 + N17 (Fig. 9A). Call types with an analyzed number greater than five were extracted for further discriminant and cluster analyses and 31 call types meet the requiment and account for 93.82% of all analyzed calls (Fig. S27). Hierarchical clustering using a between-groups linkage method that measures the squared Euclidean distance automatically grouped the 31 extracted call types into five clusters. The N17, 1 + N19, 2 + N18 and 3 + N17 call types were grouped into one cluster, and iN13 and iN15 were grouped together (Fig. 9B). Most of the call types with an IPPI median of 10 ms were grouped together, and those with an IPPI median of 9 ms were grouped together (Fig. 9B).

Figure 9 Scatterplot using the canonical discriminant function (A) and dendrogram using the hierarchical clustering method (B) of 31 extracted call types.

The “Rescaled distance cluster combine” axis in B shows the distance at which the clusters combine. When creating a dendrogram, SPSS rescales the actual distance between the cases to fall into a 0–25 unit range; thus, the last merging step to a one-cluster solution occurs at a distance of 25.

Call occurrence patterns

Almost all call types with median IPPIs of 9 ms for the last section (i.e., call types with median IPPIs of 9 ms except the N9 call type) were only observed from June 18 to 20, 2014 (Fig. 10). Most of the call types with median IPPIs of 10 ms for the last section (88%, 29 out of 33), except 1 + N10, (1 − )2 + N10, 1 + 2 + N10, and (1 − )3 + N10, were only observed from May 26 to June 4 and June 21 to 22, 2014 (Fig. 10).

Figure 10 Occurrence pattern of the 66 call types during passive acoustic monitoring periods.

Yellow patches in the matrix indicate the corresponding call types (x-axis) observed on that day (y-axis). Call types are clustered according to their median IPPI and the number on the y-axis corresponds to the call type sequence in Table 1.

Characteristics of call trains

Of the 52 extracted call trains, the estimated inter-call interval was 1.88 ± 0.39 ms (median ± QD; P5–P95: 1.05–3.04 ms, n = 278).

Discussion

Fish sonic muscles are the fastest-contracting vertebrate muscles (Rome & Lindstedt, 1998). Many soniferous fishes produce species-specific sounds by driving their swim bladders with the highly specialized sonic muscles during courtship to aggregate males and females and facilitate successful mating, especially at night and/or in highly turbid water (Fine & Parmentier, 2015; Tavolga, 1964). The spawning-related sounds produced by soniferous fishes have been widely used to identify the timing of spawning and map the areas where spawning occurs (Locascio & Mann, 2011; Turnure, Grothues & Able, 2015). The sound recording period in our study was during the spawning seasons of a majority of the local fishes because their reproduction behavior was most evident from March through June in the Pearl River Estuary (Sadovy, 1998). The spawning activity of the greyfin croaker (Pennahia anea) occurred from March–April to June (Tuuli, De Mitcheson & Liu, 2011), the spawning season of the spiny-head croaker (Collichthys lucidus) began in March and lasted until December, and the season for Belanger’s croaker (Johnius belangerii) was from April to December (Li, Chen & Sun, 2000; Sadovy, 1998).

In the present study, presumably spawning choruses were recorded daily, indicating that the sound recording location is a spawning place for local soniferous fish. The smallest inter-pulsepeak interval in our study was 8.32 ms, which was longer than and further validated the conservatively defined 8 ms pulse duration.

In this study, the call types were categorized primarily by their IPPI patterns rather than the IPPI ranges. Although there was some overlap in the range of IPPIs, N9 and N10 (A4 and B4 in Fig. 4 and Fig. S28) and iN13 and iN15 (A4 and B4 in Fig. 5) were separated based on the distribution pattern of their IPPIs.

Sound comparison of soniferous fish in the PRE

The South China Sea, with at least 2,321 fish species belonging to 35 orders, 236 families and 822 genera (Ma et al., 2008), has long been recognized as a global center of marine tropical biodiversity (Barber et al., 2000) and is one of the richest areas in China, even globally, in terms of its marine fish diversity (Huang, 1994; Ma et al., 2008). More than 834 fish species belonging to 25 orders, 124 families and 390 genera were recorded in the waters near Hong Kong (Ni & Kwok, 1999).

Comparisons with Sciaenid sounds

Fishes of the family Sciaenidae, which are commonly known as croakers or drums, are some of the most well-studied soniferous fish species, and more than 23 species in this family were recorded in the waters near Hong Kong (Ni & Kwok, 1999).

Voluntary sounds

In free-ranging conditions, big-snout croaker (J. macrorhynus) can emit voluntary purr signals with the first and the remaining IPPIs averaging 40.1 ms and 9.7 ms in the field and 35.3 ms and 10.4 ms in a large aquarium, respectively (Table 5) (Lin, Mok & Huang, 2007), which resembles the 1 + N10 call type in our study (Table 4, Fig. 6A) (note that the IPPI was equal to the summation of the pulse duration and the inter-pulse interval in Lin, Mok & Huang, 2007). In addition, the peak frequency of the pulses in 1 + N10 (mean ± sd: 1,077 ± 244, N = 1,507) was intermediate between those in the pulses of big-snout croaker purr signals as recorded in the field (mean ± sd: 1,146 ± 131, N = 250) and in a large aquarium (mean ± sd: 1,050 ± 84, N = 60). Additionally, the voluntary dual-knock signal of big-snout croaker with an average IPPI of 36.7 ms and 39.4 ms as recorded in the field and in a large aquarium, respectively (Table 5) (Lin, Mok & Huang, 2007), resembled the 1 + 1 call type in our study with an IPPI of 40.70 ± 4.08 (mean ± sd) (Table S1, Fig. S1B). These matches were further supported by the fact that the peak frequency of the pulses in the 1 + 1 call type (mean ± sd: 1077.75 ± 219.58, N = 126) was close to that of the dual-knock recorded in the field (mean ± sd: 1,133 ± 119, N = 40) or a large aquarium (mean ± sd: 1,135 ± 85, N = 50).

Table 5 Frequency and inter-pulsepeak interval (IPPI) characteristics of soniferous fish in the Pearl River Estuary.

Family	Species	Latin name	Condition	Peak frequency	IPPI	First IPPI	Last IPPI	No. signal	Comments	Reference	
Sciaenidae	Belanger’s croaker	Johnius belangerii	Voluntary	500–1,000 Hza		40 ms	20 mse			Pilleri, Kraus & Gihr (1982)	
				750–1,250 Hz					Long burst	Pilleri, Kraus & Gihr (1982)	
			Disturbance	584 ± 181 Hz	12.9 ms	14.4 ms	16.9 ms	200		Mok, Lin & Tsai (2011)	
	Big-snout croaker	J. macrorhynus	Voluntary	1,146 ± 131 Hz		40.1 ms	9.7 mse	40	Purr signalsc	Lin, Mok & Huang (2007)	
			Voluntary	1050 ± 84 Hz		35.3 ms	10.4 mse	40	Purr signald	Lin, Mok & Huang (2007)	
			Voluntary	1,133 ± 119 Hz	36.7 ms			15	Dual-knocksc	Lin, Mok & Huang (2007)	
			Voluntary	1,135 ± 85 Hz	39.4 ms			15	Dual-knocksd	Lin, Mok & Huang (2007)	
			Disturbance	808 ± 142 Hz		22.2 ms	9.5 mse	40	Purr signals	Lin, Mok & Huang (2007)	
			Disturbance	807 ± 143 Hz	10.1	22.2 ms	10.5 ms	85		Mok, Lin & Tsai (2011)	
			Disturbance	425.9 ± 93.7 Hz		19.2 ± 7.3 ms		352	Male + female	Huang (2016)	
			Disturbance	450.9 ± 106.1 Hz		20.5 ± 8.2 ms		210	Male	Huang (2016)	
			Disturbance	386.5 ± 57.1 Hz		8.0 ± 1.4 ms		142	Female	Huang (2016)	
		J. sp.	Disturbance	454.0 ± 33.7 Hz		12.8 ± 6.4 ms		28	Male + female	Huang (2016)	
			Disturbance	454.0 ± 33.7 Hz		10.6 ± 1.8 ms		25	Male	Huang (2016)	
			Disturbance	2249.9 ± 584.6 Hz		22.6 ± 10.5 ms		5	Female	Huang (2016)	
	Sciaenidae	J. distincus	Disturbance	839 ± 144 Hz		9.97 ± 0.72 ms	12.36 ± 0.53 ms		Male	Tsai (2009)	
			Disturbance	581 ± 66 Hz		10.12 ± 0.82 ms	12.53 ± 0.79 ms	210	Female	Tsai (2009)	
			Disturbance		10.8 ms	11.1 ms	12.3 ms	242		Mok, Lin & Tsai (2011)	
			Disturbance	392.4 ± 100.0 Hz		13.4 ± 4.8 ms		524	Male + female	Huang (2016)	
			Disturbance	398.1 ± 94.0 Hz		14.3 ± 2.3 ms		273	Male	Huang (2016)	
			Disturbance	352.1 ± 84.2 Hz		11.6 ± 2.7 ms		183	Female	Huang (2016)	
		J.amblycephalus	Disturbance	367.1 ± 100.8 Hz		14.5 ± 3.6 ms		58		Huang (2016)	
	Sin croaker	J. dussumieri	Disturbance	517 Hz		11.4 ms	14.9 ms			Tsai (2009)	
	White croaker	Pennahia argentata	Voluntary	457 Hz					Male	Ramcharitar, Gannon & Popper (2006)	
			Voluntary	267 Hz					Female	Ramcharitar, Gannon & Popper (2006)	
			Disturbance	543 ± 98 Hz	22.9 ms	24.0 ms	37.9 ms	104		Mok, Lin & Tsai (2011)	
			Disturbance	348.6 ± 18.1 Hz		9.4 ± 0.3 ms		23	Female	Huang (2016)	
	Greyfin croaker	P. anea	Disturbance	736 ± 115 Hz	10.6 ms	9.1 ms	12.1 ms	90		Mok, Lin & Tsai (2011)	
			Disturbance	551.9 ± 27.7 Hz		10.9 ± 1.6 ms		15	Female	Huang (2016)	
	Bighead white croaker	P. macrocephalus	Disturbance	576 ± 93 Hz	34.6 m	25.2 ms	38.1 ms	92		Mok, Lin & Tsai (2011)	
			Disturbance	425.9 ± 93.7 Hz		19.2 ± 7.3 ms		352	Male + female	Huang (2016)	
			Disturbance	450.9 ± 106.1 Hz		20.5 ± 8.2 ms		210	Male	Huang (2016)	
			Disturbance	386.5 ± 57.1 Hz		8.0 ± 1.4 ms		142	Female	Huang (2016)	
	Pawak croaker	P. pawak	Disturbance	736 ± 101 Hz	9.1 ms	8.5 ms	9.7 ms	169		Mok, Lin & Tsai (2011)	
			Disturbance	388.1 ± 41.6 Hz		11.2 ± 2.1 ms		15	Female	Huang (2016)	
	Large yellow croaker	Pseudosciaena crocea	Voluntary	550–750 Hza				182	Single pulse	Liu, Xu & Qin (2010)	
			Voluntary	800–850 Hza	90–150 msa				2–3 pulse signal	Ren et al. (2007)	
			Disturbance	800–850 Hza	>30 msa				2–5 pulse signal	Liu, Xu & Qin (2010)	
			Disturbance	264.7 ± 22.3 Hz		11.5 ± 3.1 ms		29	Female	Huang (2016)	
	Southern meagre	Argyrosomus japonicas	Voluntary	686 ± 203 Hz	24 ± 3 ms			210	Male	Ueng, Huang & Mok (2007)	
			Voluntary	587 ± 190 Hz	23 ± 3 ms			164	Female	Ueng, Huang & Mok (2007)	
	Yellow Drum	Nibea albiflora	Voluntary	650 ± 20 Hz						Ren et al. (2007)	
			Disturbance	293.1 ± 56.4 Hz		12.2 ± 2.2 ms		23		Huang (2016)	
	Reeve’s croaker	N. acuta	Voluntary	630 ± 15 Hz						Ren et al. (2007)	
			Disturbance	<500 Hza						Tsai (2009)	
	Tiger-toothed croaker	Otolithes ruber	Disturbance	354–1,717 Hza	8.3–12.2 msa			17		Mok, Lin & Tsai (2011)	
	Blackmouth croaker	Atrobucca nibe	Disturbance		47.0–57.8 msa			1		Mok, Lin & Tsai (2011)	
Trichiuridae	Cutlassfish	Trichiurus haumela	Voluntary	628 ± 11 Hz						Ren et al. (2007)	
Pristigasteridae	Elongate ilisha	Ilisha elongata	Voluntary	251 ± 18 Hz						Ren et al. (2007)	
Ariidae	Sea catfish	Arius sp.	Voluntary	735 ± 12 Hz						Ren et al. (2007)	
		A. maculates	Disturbance		0.47–4.33 msa,b				5–11 pulse signal	Mok, Lin & Tsai (2011)	
Glaucosomatidae	Pearl perch	Glaucosoma buergeri	Disturbance		30 ms				2–9 pulse signal	Mok et al. (2011)	
Priacanthidae	Bigeye snapper	Priacanthus macracanthus	Disturbance	172 Hz	15.9 ms					Tsai (2009)	
Terapontidae	Trumpeter perch	Pelates quadrilineatus	Disturbance	690 ± 171 Hz	4 ms					Tsai (2009)	
Haemulidae	Javelin grunter	Pomadasys kaakan	Disturbance		94.1 ms					Tsai (2009)	
Notes.

Except when mentioned, the results are given as the mean or mean ± standard deviation (sd).

a denotes results given in a range.

b denotes results given for the inter-pulse interval.

c denotes results recorded in the field.

d denotes results recorded in a large aquarium.

e denotes results that are the mean of all the IPPIs except the first IPPI.

It is possible that J. macrorhynchus might emit dual-knock and purr signals in series and creates a multiple section call type, such as one dual knock combined with one purr which may result in a synthetic three section call type of 1 + 2 + N10 (time gap between the two signals was equal to 10 ms) or a four section call type of 1 + 1 + 1 + N10 (time gap between the two signals was over 20 ms). However, both of the synthetic 1 + 2 + N10 and 1 + 1 + 1 + N10 signals with the third IPPI ascribed to the first IPPI of the purr signal and averaged at 40.1ms (Lin, Mok & Huang, 2007) cannot match either the 1 + 2 + N10 or the 1 + 1 + 1 + N10 call types in our study, since both of which with the third IPPI of less than 30 ms (Fig. S7A and Fig. S12B). Belanger’s croaker can emit sounds with the first IPPI much longer than subsequent IPPIs, which follow at regular intervals of approximately 20 ms (Pilleri, Kraus & Gihr, 1982) and resemble the 1 + N19 call type in our study, although the first IPPI in Belanger’s croaker (approximately 40 ms) (Table 5) (Pilleri, Kraus & Gihr, 1982) was smaller than that in the 1 + N19 call type (median at 71.36 ms) (Table 4, Fig. 6C). Their similarity was further strengthened by the fact that the temporal and frequency characteristics of the signal emitted by Belanger’s croaker, which consists of 4–14 pulses with a 140–260 ms call duration, a 500–1,000 Hz peak frequency and a majority of the energy within the 500–4,000 Hz frequency band (Pilleri, Kraus & Gihr, 1982), resemble those of the 1 + N19 call type, which consists of 3–12 pulses with a 97.37–272.85 ms call duration and peak frequency median of approximately 789 Hz (Table 4).

Sounds from the white croaker (Pennahia argentata) (Ramcharitar, Gannon & Popper, 2006; Takemura, Takita & Mizue, 1978), southern meagre (Argyrosomus japonicus) (Ueng, Huang & Mok, 2007), yellow drum (Nibea albiflora) (Ramcharitar, Gannon & Popper, 2006; Ren et al., 2007; Takemura, Takita & Mizue, 1978), Reeve’s croaker (N. acuta or Chrysochir aureus) (Ren et al., 2007; Trewavas, 1971) and large yellow croaker (Liu, Xu & Qin, 2010; Ren et al., 2007) were also compared. However, these sounds (Table 5) did not match any call types in our study based on their temporal and/or frequency characteristics.

Belanger’s croaker can also emit long bursts with a peak frequency of 750–1,250 Hz (Pilleri, Kraus & Gihr, 1982), and a chorus sound of unknown species recorded in Xiamen Harbor of East China Sea from 1981 to 1982 with sound energy concentrated in the 700–1,600 Hz frequency band and a peak frequency of 1,250 Hz was proposed to be emitted by Belanger’s croaker (Zhang et al., 1984). Chorus sounds of the genus Johnius (possibly J. fasciatus or J. amblycephalus) and the genus Pennahia (possibly P. miichthioides) recorded in the Bohai Sea and Yellow Sea from 1989–1990 were also reported. The sounds emitted by the former genus have an average peak frequency of 2,000 Hz and a majority of energy concentrated in the 1,000–4,000 Hz frequency band, whereas the sounds emitted by the latter genus have an average peak frequency of 400 Hz and majority of energy concentrated in the 200–800 Hz frequency band (Xu & Qi, 1999). Chorus sounds of the spiny-head croaker were recorded in the South China Sea, with a majority of energy concentrated in the 500–1,250 Hz frequency band and a peak frequency of approximately 1,000 Hz (Qi, Zhang & Song, 1982). Chorus sounds of unknown species recorded in the adjacent waters of Xiamen Harbor of the East China Sea from 1981 to 1982, with sound energy concentrated in the 700–1,600 Hz frequency band and peak frequencies of 800 Hz and 1,000 Hz, were ascribed to the spiny-head croaker (Zhang et al., 1984). However, detailed waveform, spectrum and statistical results for the temporal and frequency characteristics of individual sounds in these choruses were not available, preventing direct comparison with our study.

Disturbance sound

Sound recorded under disturbance, e.g., under hand-held conditions is possibly not significantly different from those recorded under voluntary conditions and can be employed to match the sound in the field (Lin, Mok & Huang, 2007). In addition, the sound recording region is a hot spot of humpback dolphin (Wang et al., 2015b), the predator of soniferous fish, which may impose a stress for local fish and may trigger them to emit signal similar to the hand-held disturbance call. Thus, we also compared the disturbance sound of the sciaenid species distributed in our study region, including Belanger’s croaker (Mok, Lin & Tsai, 2011), big-snout croaker (Huang, 2016; Lin, Mok & Huang, 2007; Mok, Lin & Tsai, 2011), J. distincus, J.amblycephalus and J. sp., sin croaker (J. dussumieri), white croaker, greyfin croaker, bighead white croaker (P. macrocephalus), pawak croaker (P. pawak), Reeve’s croaker, tiger-toothed croaker (Otolithes ruber), and blackmouth croaker (Atrobucca nibe) (Huang, 2016; Mok, Lin & Tsai, 2011; Tsai, 2009). However, the temporal and frequency patterns of these signals did not match any call types in our study (Table 5).

Comparison with other soniferous fish families

Sounds from other soniferous fish families, including cutlassfish (Trichiurus haumela, family: Trichiuridae), elongate ilisha (Ilisha elongata, family: Pristigasteridae) (Ren et al., 2007), sea catfish (Arius sp. and A. maculates, family: Ariidae) (Mok, Lin & Tsai, 2011; Ren et al., 2007), pearl perch (Glaucosoma buergeri, family: Glaucosomatidae) (Mok et al., 2011), bigeye snapper (Priacanthus macracanthus, family: Priacanthidae), trumpeter perch (Pelates quadrilineatus, family: Terapontidae) and javelin grunter (Pomadasys kaakan, family: Haemulidae) (Tsai, 2009) were also compared with our call types but did not match any call types in our study in the temporal and spectral characteristics (Table 5).

Comparison with biological sounds from other passive acoustic monitoring sites

The statistical parameters of the eight types of wild fish sounds recorded in seven estuaries of the west coast of Taiwan using passive acoustics were unfortunately not available, which restricted direct comparison (Mok, Lin & Tsai, 2011). However, the general trend of the 1 + N10 and 1 + N12 call types in our study resembles their type B signal (Mok, Lin & Tsai, 2011), with the first inter-pulse interval much longer than the following ones that had a non-increasing inter-pulse interval toward the end of the call, and the N17 call type in our study resembles their type E signal (Mok, Lin & Tsai, 2011), with a gradually increasing inter-pulse interval toward the end of the call and the sound energy concentrated in discrete bands. Sounds with much longer second or third inter-pulse intervals, which resemble our 2 + N and 3 + N, respectively, were also observed in the Chosui River in Taiwan (Mok, Lin & Tsai, 2011), but the sound producer was not identified. Four call types from three recording sites on the northwestern coast of Taiwan were recorded, with the call type identical to the purr signal of J. macrorhynus dominated the soundscape and was the most abundance call type of these sites (Huang, 2016). The waveform of call type T3 resemble our call types of iN13 and iN13 (Huang, 2016).

Occurrence pattern of call types

In the field environment, to communicate without misinterpreting messages and to avoid jamming, different species of a fish community will partition the underwater acoustic environment (Ruppé et al., 2015). In our study, most call types with IPPI medians at 9 ms and 10 ms were observed at times that were exclusive from each other, suggesting they might have been produced by different species.

The spotted seatrout (Cynoscion nebulosus) is one of the few sciaenid species that produces as many as four types of call (Mok & Gilmore, 1983). It is likely that most sciaenid species have fewer call types. Of all the 66 call types recognized in the survey sites, some of the which might come from the same species. According to the result of cluster analysis, five clades were revealed. However, it is still too early to hypothesize that these groups belong to the call repertoire of five species. Additional studies with more controlled conditions, such as in an aquarium or with field recording equipped with a high-definition sonar system such as the DIDSON Dual-frequency Identification Sonar system, will be required to identify the species producing the calls in our study.

Call trains

Due to the relative simplicity of vocal mechanisms and lack of ability to produce complex calls, fish typically emit sounds with variation in either the temporal and/or frequency patterning (Rice & Bass, 2009). As most of the call types were identified based on the number of sections and the repetition of the anterior section, it is likely that a species might be able to produce several call types by varying the anterior sections of the call as a response to the variable external stimuli. Additionally, the temporal and spectral characteristics of fish signals are involved in information coding and are important parameters for the recognition of sound in fishes (Malavasi, Collatuzzo & Torricelli, 2008; Spanier, 1979). In the present study, fish sounds tended to be frequency modulated, e.g., the peak frequency of the pulses within a call were variable (Fig. 2F), and amplitude modulated, e.g., the iN13 and iN15 call types. This is possible because the amplitude of the sound is determined by the swim bladder (Fine et al., 2001; Tavolga, 1964) and the dominant frequency of the signal is determined by the sonic muscle twitch duration and the forced response of the swim bladder to sonic muscle contractions rather than the natural resonant frequency of the swim bladder (Connaughton, Fine & Taylor, 2002). Additionally, the length of the sonic muscle fibers also related to the body size of the fish (Parmentier & Fine, 2016).

Passive hearing by the dolphin

The Pearl River Estuary shelters the world’s largest known population of Indo-Pacific humpback dolphins (Chen et al., 2010; Jefferson & Smith, 2016; Preen, 2004), with an estimated population of 2,637 (Coefficient of variation of 19% to 89%) (Chen et al., 2010; Jefferson & Smith, 2016). The general preference of this species for estuarine habitats and coastal and shallow water (<30 m depth) distribution make it susceptible to the impacts of human activity (Jefferson & Smith, 2016). The current conservation status of the Chinese white dolphin meets the IUCN Red List criteria for classification as Vulnerable; however, the conservation management in a majority of its distribution range is severely inadequate, and the humpback dolphin population in the Pearl River Estuary is declining by 2.5% annually (Karczmarski et al., 2016).

The humpback dolphin appears to rely almost exclusively on fish for food (Barros, Jefferson & Parsons, 2004; Parra & Jedensjö, 2014). Its prey includes the fish families of Sciaenidae (croakers), Engraulidae (anchovies), Trichiuridae (cutlassfish), Clupeidae (sardines), Ariidae (sea catfish) and Mugilidae (mullets) (Barros, Jefferson & Parsons, 2004; Parra & Jedensjö, 2014). Notably, the majority of these species are soniferous fishes (Banner, 1972; Fish & Mowbray, 1970; Ren et al., 2007; Whitehead & Blaxter, 1989). The top three most important and frequent prey of humpback dolphins in the Pearl River Estuary are the brackish water species of croaker (Johnius sp.), spiny-head croaker (C. lucidus), and anchovies (Thryssa spp., T. dussumieri and/or T. kammalensis) (Barros, Jefferson & Parsons, 2004). The former two are soniferous fishes (Ren et al., 2007), and the latter might be capable of making sounds (Whitehead & Blaxter, 1989). Additionally, it has been proposed that dolphins rely heavily on eavesdropping (passive listening) (Barros, 1993; De Oliveira Santos et al., 2002) during the search phase of the foraging process (Gannon et al., 2005).

In addition to emitting high-frequency pulsed sounds for echolocation and navigation, humpback dolphins can produce narrow-band, frequency-modulated whistles with a fundamental frequency range of 520–33,000 Hz (Wang et al., 2013) and apparent source levels of 137.4 ± 6.9 dB re 1 µPa in rms (Wang et al., 2016) for communication. The fish sounds recorded in this study, which were characterized by a peak frequency between 500 and 2,600 Hz and a maximum zero-to-peak sound pressure level greater than 164 dB, were well within the frequency range of humpback dolphin whistles. It is highly probable that the fish sounds function as acoustic clues of prey to the dolphin, i.e., the dolphin relies heavily on passive hearing during the search phase of the foraging process. On the other hand, the brackish water species of C. lucidus and tapertail anchovy (Coilia mystus, Family: Engraulidae) were the top two predominant species in the seawater/freshwater mixing zones of the Pearl River Estuary (Zhan, 1998), accounting for 89% and 72% of the numbers and biomass, respectively, of the whole fish stock in the Pearl River Estuary region (Wang & Lin, 2006). While, the soniferous fish C. lucidus was observed to be the second-most important prey for humpback dolphin, but the non-soniferous fish C. mystus was not identified in their prey spectrum (Barros, Jefferson & Parsons, 2004). This fact can further reinforce the passive hearing mechanism of the local humpback dolphin.

Importance and application

The high biodiversity of fish fauna dwell at the Pearl River Estuary is a treasure of genetic resources and has great potential application value. However, the loss of the fishery stocks over time has been devastating. Historically poor management and overfishing of wild stocks of the large yellow croaker resulted in overwhelming collapses throughout its geographic range (Liu & Sadovy, 2008), and although substantial funds have been provided and many remedial actions such as fishery control, restocking and marine aquaculture have been applied. However, aquaculture can only supplement, rather than substitute, wild fisheries (Goldburg & Naylor, 2005). No evidence of recovery in the wild stock of large yellow croaker has been observed, and its genetic diversity continues to decrease (Liu & Sadovy, 2008). Similar lessons can be learned from the Atlantic salmon (Salmo salar) (Goldburg & Naylor, 2005). Given the sharp declines in fish stocks, especially of the larger species of croakers owing to overfishing in the Pearl River Estuary, and given that fishing pressure is still high and may be even higher in the future, management activities such as more effective fishing moratoriums should be applied to protect the remaining croakers and other fisheries during the spawning season, especially at their spawning grounds. The baseline data of the ambient biological acoustics in our study represent a first step toward mapping the spatial and temporal patterns of soniferous fishes and are helpful for the protection, management and effective utilization of fishery resources. In addition, since marine environmental impact assessment must be based upon a good understanding of the local biodiversity, the baseline data of suspected fish sounds in our study can facilitate the evaluation of the impacts from various infrastructure projects on local aquatic environments by comparing the baseline to post-construction and/or post-mitigation effort data. Additionally, there is a large body of evidence that the distribution pattern of marine mammals tends to be correlated with the spatial–temporal variability of their prey (Benoit-Bird & Au, 2003; Wang et al., 2015a; Wang et al., 2014a); this correlation was also proposed for the vulnerable local humpback dolphin (Wang et al., 2015b), and the fine-scale distribution pattern of soniferous fishes can aid in the conservation of these emblematic dolphins.

Conclusion

Using passive acoustic monitoring, the ambient biological sounds in the Pearl River Estuary were recorded and analyzed. In addition to single pulse, the sounds tend to possess a pulse train structure with a peak frequency between 500 and 2,600 Hz and most of the energy below 4,000 Hz. Sixty-six call types were identified based on the number of sections, temporal characteristics and amplitude modulation patterns. Most of the call types with IPPI medians at 9 ms and those with medians at 10 ms were observed at times that were exclusive from each other, suggesting that they might be produced by different species. A literature review suggested that the 1 + 1 and 1 + N10 call types might belong to big-snout croaker (J. macrorhynus) and 1 + N19 might be produced by Belanger’s croaker (J. belangerii). The baseline data of suspected fish sounds in our study can facilitate the evaluation of the impact from various infrastructure projects on the local aquatic environments by comparing the baseline to post-construction and/or post-mitigation effort data, and the fine-scale distribution pattern of soniferous fishes can aid in the conservation of the local vulnerable humpback dolphins.

Supplemental Information

Supplemental Information 1 Supplemental figures

Click here for additional data file.

Supplemental Information 2 Supplemental tables

Click here for additional data file.

We gratefully acknowledge Wenjun Xu of the Ningbo No. 2 High School in Zhejiang Province for her statistical assistance. Special thanks are also extended to Andrew J. Read of the Duke University Marine Laboratory for his helpful discussion about this study.

Additional Information and Declarations

Competing Interests

Author Contributions

Animal Ethics

Data Availability

Guo-Qin Duan and Han-Jiang Cao are employees of Hong Kong-Zhuhai-Macao Bridge Authority, Guangzhou, P.R. China.

Zhi-Tao Wang conceived and designed the experiments, performed the experiments, analyzed the data, contributed reagents/materials/analysis tools, wrote the paper, prepared figures and/or tables, reviewed drafts of the paper.

Douglas P. Nowacek contributed reagents/materials/analysis tools, wrote the paper, reviewed drafts of the paper.

Tomonari Akamatsu wrote the paper, reviewed drafts of the paper.

Ke-Xiong Wang conceived and designed the experiments, performed the experiments, wrote the paper, reviewed drafts of the paper.

Jian-Chang Liu, Guo-Qin Duan and Han-Jiang Cao reviewed drafts of the paper.

Ding Wang conceived and designed the experiments, reviewed drafts of the paper.

The following information was supplied relating to ethical approvals (i.e., approving body and any reference numbers):

Permission to conduct the study was granted by the Ministry of Science and Technology of the People’s Republic of China. The research permit was issued to the Institute of Hydrobiology of the Chinese Academy of Sciences (Permit number: 2011BAG07B05).

The following information was supplied regarding data availability:

Wang, Zhitao (2017): Fish acoustics in China. figshare.

http://dx.doi.org/10.6084/m9.figshare.5001353.v1.

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
