# Peer review of "Diversity of fish sound types in the Pearl River Estuary, China"

_PeerJ, doi:10.7717/peerj.3924_

## Round 0.1 · original submission · Major Revisions

· Academic Editor

Major Revisions

We have had real issues getting reviewers to look at this paper. This is partly because of the vast number of figures and lack of synthesis. You really need to reduce the number of figures to something more sensible and use them to highlight key points. At the moment the reader is baffled by detail (lots of them could be supplementary. The paper is also lacking in direction in some places. It considers cetaceans, soniferous fish noises and fisheries management, Really it needs to focus on the central theme of sounds produced by fish. The other aspects are interesting footnotes worthy of brief mention in the discussion.

Reviewer 1 ·

Basic reporting

No comment

Experimental design

No comment

Validity of the findings

No comment

Additional comments

General suggestions:
1.In Result section, the authors mentioned 23 sciaenid species had been recorded in Hong Kong waters by citing a 1999 paper. Information about the sciaenid species occurring in the survey area in more recent years, if available, is useful to help confirm the caller’s identity. The voluntary calls of J. macrorhynchus and J. belangerii match some of the call types, it is expected that these two species should present in the study site.
2.The authors should give the relative abundance of each call type occurring in each recording day (or in each of the two continuous recording session) if possible. This information is helpful to interpret number of soniferous species in the area that produced these 66 call types.
3.Please aware that no vertical lines should be used in tables.
4.All characteristics of 3-8 section calls are presented in the supplementary section. It is suggested to add one figure following Fig. 7 to illustrate three of these call types.
5.Comparisons were made among the call types recognized in this study with the hand-held disturbance sounds of several sciaenid species published in the literature and no match was noticed except for Johnius macrorhynchus. Despite of the similarity among the disturbance and voluntary call types in J. macrorhynchus (Lin et al., 2007), this condition may not be a general rule for other sciaenid species. Hand-held disturbance call might be a vocal response toward the highest stress to a fish and this high stressful condition may not be the one causing the voluntary calls heard in the survey site. The purrs emitted when the fish was confined in a small tank, the repeating cycle reduced (with a longer pulse period).
Johnius macrorhynchus emits two call types (Lin et al., 2007), the dual knocks and the purr. It is possible that J. macrorhynchus emit these two types in series and creates a multiple section call type.
6.The spotted seatrout (Cynoscion nebulosus) is one of the few sciaenid species that produces as many as four types of call (Mok, H.k. and R.G. Gilmore. 1983. Analysis of sound production in estuarine aggregations of Pogonias cromis, Bairdiella chrysoura, and Cynoscion nebulosus (Sciaenidae). Bulletin of the Institute of Zoology, Academia Sinica. 22(2): 157-186.). It is likely that most sciaenid species have fewer call types. Because there are 66 call types recognized in the survey sites, one can guess some of the call types must come from a single species.
As most of the call types were identified based on the number of sections and the repetition of the anterior section, it is likely that a species might be able to produce several call types by varying the anterior sections of the call as a response to the variable external stimuli.
According to the result of cluster analysis, five clades were revealed. Can the authors hypothesize that these groups belong to the call repertoire of five species?
7.Spiny croaker is a small croaker species, its call type could be distinct to other related species due to it small sonic muscles.
8.Huang (Huang Po-Wei. 2016. Sound characteristics and spatiotemporal variability of Scienidae in the coastal water of northwestern Taiwan. Master Thesis, National Dong Hwa University, 109pp.) recognized only four call types from three recording sites on the northwestern coast of Taiwan where 9 sciaenid species have been recorded (Johnius amblycephalus, J. distinctus, Lamichthys crocea, Nibea althiflora, Pennahia anea, P. argentata, P. microcephalus, P. pawak, and a undescribed Johnius sp. which has been has been misidentified as J. macrorhynchus. The call type is identical to the purr call of ‘J. macrorhynchus’ was the most abundance type at these sites. The soundscape was dominated by only one sound type.
9.Please explain how the order of the call types in Fig. 8 was decided?

Specific suggestions:
1.Line 22, ‘species-specific sound’ should be use cautiously because sounds of some soniferous fishes might to so similar that they don’t have a species-specific call type.
2.Line 70, : ‘sp.’ should be changed into ‘sp.’ (in italics)
3.Line 71: : ‘spp.’ should be changed into ‘spp.’(in italics)
4.Line 88: ‘Nibea diacanthus’ should be changed to ‘Protonibea diacanthus’.
5.Line 101:’ the spatial and temporal patterns of soniferous fishes, should consider to be changed into ‘the spatial and temporal distribution patterns of soniferous fishes in the estuary’.
6.Line 102: ‘of the damage to aquatic environments’ may consider to be changed into ‘of the damage to aquatic environments (e.g., spawning grounds of the sciaenids)’.
7.Lines 202-205”… and the call duration was dependent on the number of pulses in a call these parameters were not included in the PCA, Canonical discriminant analysis and hierarchical cluster analyses.” However, pulse number in a call was a parameter valuable in matching call type 1+N19 with the known voluntary call of Belanger’s croaker. As such call duration remains a useful taxonomic character.
8.In line 261-262, ‘Hierarchical clustering using a between group method that measure the squared Euclidean distance automatically group the 31 extracted call types…’. Please distinguish ‘the 31 extracted call types’ with the 66 call types mentioned elsewhere in the text.
9.Line 352: ’underhand held condition’ should be changed into ‘under hand-held condition’.
10.Line 501: ‘chinese ‘ should be changed into ‘Chinese’.
11.Line 504 : ‘spp.’ should be changed into ‘spp.’ (in italics)
12.Line 364: ‘sp.’ should be changed into ‘sp.’. *(in italics)
13.Line 12: ‘Soundfrom’ should be changed into ‘Sound from’.
14.Line 362: ‘Sound from other soniferous families, including cutlassfish (Trichiurus haumela, family Trichiuridae)(Ren et al., 2007)’. Because the subject of Ren et al. (2007) paper is about the sound of large yellow croaker (not about the cutlassfish sound), I suggest the authors should recheck if cutlassfish is actually soniferous.
15.Species names in the references should be printed in italics.
16. Lines 681, 692, 701, 702, 704, a full stop should be put behind the table number.

Reviewer 2 ·

Basic reporting

see below

Experimental design

see below

Validity of the findings

see below

Additional comments

The sheer number of figures (often multiple panel) and tables as well as non standard nomenclature to describe the sounds, prevents me from investing the time to adequately review this manuscript.

I suggest the authors give serious thought about what and is not necessary as most of the figures are unneeded even as supplemental files and revise the manuscript to length that is more fitting its overall content

---

## Round 0.2 · Minor Revisions

· Academic Editor

Minor Revisions

Please accept my sincere apologies for the late response to this paper - I have spent much of hte summer on a remote island off the coast of Malaysia with unexpectedly terrible internet access.

Please pay close attention to the suggestions made the the reviewer. I also remain firmly of the opinion that you should be much more succinct in your linking of your research to the plight of the humback dolphin - it is a valid point to make but the focus of your paper is on the sounds emitted by fish. The correspondance between the sounds produced by fish and the sensitivity of endangered humbacked dolphins could be a whole and significant paper in itself (please provide the scientific name of the dolphin). Using the characteristics of the sounds produced by fish to map the distrbution of preferred prey and the known distribution of the dolphin would ba a fabulous paper.

Reviewer 1 ·

Basic reporting

No comment

Experimental design

No comment

Validity of the findings

No comment

Additional comments

1. Line 49: I suggest that the authors should consider changing the word ‘Most’ into ‘Some’.

2. Line 60, ‘Pseudoscianea crocea’ should be changed to ‘Larimichthys crocea’.

3. Line 71: ‘(signature)’ is suggested to be deleted.

4. Line 87: ‘water proof’ changed to ‘waterproof’.

5. Line 115: ‘Total call duration’ changed to ‘Call duration’.

6. Line 150: ‘Fig. 4-5’ should be changed to ‘Figs. 4-6’.

7. It is suggested that (1) at least one representative call type of each call category differing in number of sections should be included in figures 4-6 so that they can be cited in the text (i.e., putting at least one representative call type for 1-8 section calls in these three figures); (2) some redundancy or highly resembling call types original placed in these three figures should be reduced; (3) figures and tables in the supplementary section should not be cited in the text. If these suggestions are accepted, then (Fig. S1-S26) in line 150 should be deleted.

8. Line 178: ‘Ethical statement’, I don’t think this study involves ethical issue with the animals surveyed as only the biological sounds in the field were recorded by an noninvasive method.

9. Line 230-231: “Call types with an analyzed number greater than five were extracted for further discriminant and cluster analyses and 31 call types meet the requirement.” Does it mean that these 31 types were more common than the other 35 call types in this study site? If this is true, then it should be stated clearly in the Occurrence pattern of call types in Discussion (line 353 or other more appropriately part of the text) to give an idea about the relative abundance of the 66 types. Also see comment 17.

10. Line 268: ‘with Sciaenidae sounds’ should be changed to ‘with sciaenid sounds’

11. Line 327: ‘of the species’ should be changed to ‘of the sciaenid species’ and Line 329: ‘Sciaenidae’ should be deleted.

12. Line 335: ‘Arius sp.’ should be changed into ‘ Arius sp.’ (‘sp.’ following a generic name should not be in italic type.)

13. Line 341: ‘Comparison with other passive acoustic monitoring sounds’ is suggested to be changed to ‘Comparison with biological sounds from other passive acoustic monitoring sites’

14. Line 360: ‘come from a single species’ changed to ‘come from the same species’.

15. The problem with peak frequency was discussed between Line 375-378, as length of the sonic muscle fibers with also related to body size, this factor should be mentioned as well.
The book chapter by Parmentier, E and M.L. Fine., entitled, ‘Fish sound production: insights.’ (In R.A. Suthers et al (eds.). 2016. Vertebrate Sound Production and Acoustic Communication, Springer Handbook of Auditory Research 53, Springer International Publishing Switzerland, DOI 10.1007/978-3-319-27721-9_2) should be cited.

16. Line 637: “inter-pulsepeak interval’ differs from ‘inter pulse peak interval’ used in labelling the X-axis of Fig.4 B. Please check.

17. There are too many insets within Figures 4-6. For example (1) Characteristics of the (A) N9, (B) N10, (C) N13, (D) N17 call types were given. The authors should consider leaving just N9 and N17 in this figure. (2) The insets on regression of call duration and number of pulses in a call for each call types are not necessary.

18. In Table 1, ‘Typ and e’ in the upper left corner of the table are separated ; they should be together.

---

## Round 0.3 · accepted · Accept

· Academic Editor

Accept

I think this manuscript is now at a stage where, with a few very minor modifications (which can be addressed in production), it can be accepted for publication. I have removed one superfluous sentence and suggested that you reword another in the discussion.